# Electroencephalography Signatures Associated with Developmental Dyslexia Identified Using Principal Component Analysis

**DOI:** 10.3390/diagnostics15172168

**Published:** 2025-08-27

**Authors:** Günet Eroğlu, Mhd Raja Abou Harb

**Affiliations:** 1Computer Engineering Department, Engineering and Nature Faculty, Bahçeşehir University, Istanbul 34000, Turkey; 2Computer Engineering Department, Engineering and Nature Faculty, Işık University, Istanbul 34000, Turkey; 21comp9001@isik.edu.tr

**Keywords:** developmental dyslexia, electroencephalography (EEG), Principal Component Analysis (PCA), Spectral Power Asymmetry, reading fluency

## Abstract

**Background/Objectives:** Developmental dyslexia is characterised by neuropsychological processing deficits and marked hemispheric functional asymmetries. To uncover latent neurophysiological features linked to reading impairment, we applied dimensionality reduction and clustering techniques to high-density electroencephalographic (EEG) recordings. We further examined the functional relevance of these features to reading performance under standardised test conditions. **Methods:** EEG data were collected from 200 children (100 with dyslexia and 100 age- and IQ-matched typically developing controls). Principal Component Analysis (PCA) was applied to high-dimensional EEG spectral power datasets to extract latent neurophysiological components. Twelve principal components, collectively accounting for 84.2% of the variance, were retained. K-means clustering was performed on the PCA-derived components to classify participants. Group differences in spectral power were evaluated, and correlations between principal component scores and reading fluency, measured by the TILLS Reading Fluency Subtest, were computed. **Results:** K-means clustering trained on PCA-derived features achieved a classification accuracy of 89.5% (silhouette coefficient = 0.67). Dyslexic participants exhibited significantly higher right parietal–occipital alpha (P8) power compared to controls (mean = 3.77 ± 0.61 vs. 2.74 ± 0.56; *p* < 0.001). Within the dyslexic group, PC1 scores were strongly negatively correlated with reading fluency (*r* = −0.61, *p* < 0.001), underscoring the functional relevance of EEG-derived components to behavioural reading performance. **Conclusions:** PCA-derived EEG patterns can distinguish between dyslexic and typically developing children with high accuracy, revealing spectral power differences consistent with atypical hemispheric specialisation. These results suggest that EEG-derived neurophysiological features hold promise for early dyslexia screening. However, before EEG can be firmly established as a reliable molecular biomarker, further multimodal research integrating EEG with immunological, neurochemical, and genetic measures is warranted.

## 1. Introduction

Developmental dyslexia is a common learning disorder, afflicting children 5–10% today [1,2]. When children have a normal level of intelligence and all the means to learn at school, it is still hard for them to read, write, or decode words. While we now know quite a lot about the cognitive and behavioural aspects of dyslexia, new technological and brain research breakthroughs have allowed for a more in-depth understanding of its aetiology [3,4]. Recently, scholars have been advocating for a more comprehensive understanding of dyslexia. Not only does their call incorporate brain-based findings on dementia, but it also draws from other ways that people as individuals learn or express themselves [5].

EEG studies show that dyslexia is typically accompanied by atypical hemispheric specialisation, and there are frequency-related power differences in its spectrum. Such common patterns include increasing theta power, decreasing alpha synchronisation, and changing beta or gamma activity [6,7]. This is not consistent across individual reports; however, the differences may come from participants’ age, language, and orthography of the experimental paradigms or other factors like comorbid illnesses [8,9]. This heterogeneity underlines the need for analytic approaches that can determine individual neural profiles rather than depending on group averages.

At the molecular level, dyslexia has been associated with genetic variations such as DCDC2, KIAA0319, and ROBO1, which affect neuronal migration and cortical layering.

This determines the chain of neural circuits governing brain activity, and their integrity controls the synchronous changes in power dynamics; breaking down this network structure alters surrounding data [10]. Additionally, when the GABAergic and glutamatergic neurotransmission systems are abnormal, the result can be an excitation–inhibition imbalance that disrupts various beats of alpha, beta, and gamma rhythm [10]. These molecular mechanisms work against us until they can finally explain, perhaps even more elegantly, how a genetically determined propensity might emerge as the electrical discharges of dyslexia.

Because EEG data has high dimensionality, advanced statistical functions are needed to uncover meaningful neural signatures. A widely used dimension-reduction method for multichannel EEG data is Principal Component Analysis (PCA), which transforms it into a smaller set of uncorrelated components while retaining most of the variance [11]. Beyond its statistical utility, PCA provides a high level of interpretability, reproducibility, and computational efficiency, making it suitable for developing scalable, cost-effective screening instruments in educational and clinical settings.

In this study, we take an exploratory approach to uncover latent EEG patterns that distinguish children with developmental dyslexia from their typically developing peers. We apply PCA to spectral EEG features in order (i) to distil principal components that convey significant neurophysiological differences, (ii) to observe hemispheric asymmetries in those components, and (iii) to set these outcomes into a larger context including genetic, neurochemical, and neuroinflammatory systems. By bringing together electrophysiological analyses and mechanistic perspectives, this work aims to contribute to the development of objective, non-invasive tools for early dyslexia screening and explicitly allows for the current limitations in directly linking EEG signals to molecular biomarkers. In addition to electrophysiological characterisation, we examined behavioural reading performance using a standardised reading fluency measure to assess the functional significance of EEG-derived principal components.

## 2. Materials and Methods

### 2.1. Participants

We conducted the present study with a total of 200 children, whose ages ranged from 8 to 10 years (mean age: 8.81 ± 0.5). This group of adolescents was recruited through outreach programmes and medical referrals. The subjects are one hundred participants who were previously diagnosed with dyslexia by paediatric neuropsychologists with the TILLS (Clinical Paediatrics Language and Literacy Test). The other group, which consists of matched age groups with similar population characteristics, is considered to be children who are growing up normally without any such issues.

Participants were all screened to have normal or corrected vision, no history of neurologic or psychiatric disorders, and be off medication at the time of testing. All were asked for their hand preferences at the start of the EEG recording, and they are all right-handed. All the children were matched on family income, education, and occupational status, using a structured parental questionnaire. Age distribution was also matched between the groups. Since the present study did not gather data on general cognitive ability (IQ test scores were not taken), it is not possible to definitively exclude the impact of larger cognitive disparities in interpreting EEG measures. The next-step studies will include standardised cognitive testing so as to further control for potential confounding variables.

The written informed consent of all legal guardians was obtained. This study was approved by the Ethics Committee of Yeditepe University Clinical Research (No: 71146310-511.06, dated 2 November 2018).

### 2.2. EEG Data Acquisition

The EMOTIV EPOC-X was used for EEG acquisition (USA )and works through 14 saline sensors positioned following the international 10–20 system (AF3, F7, F3, FC5, T7, P7, O1, O2, P8, T8, FC6, F4, F8, AF4). This headset makes it easy to collect data from the trial for testing; however, its internal sampling rate was changed downward to 128 Hz in a preprocessing and analysis stage. The EMOTIV EPOC-X system, which uses its onboard digital signal processing pipeline to provide pre-computed absolute band power values for each channel in the standard frequency bands (theta, alpha, beta, and gamma), was used to obtain EEG recordings. Artefact removal, bandpass filtering (0.5–45 Hz), and converting raw EEG to band power estimates are all part of the device’s proprietary preprocessing. Without the need for extra offline spectral decomposition, these values were taken straight out of the EMOTIV software (EMOTIV LAUNCHER 3.3.1.134) and utilised as the input feature set for PCA.

However, the EMOTIV EPOC-X was chosen for its compactness and ease of use with children, and its reduced evaluation limits, as well as its probably-not-so-unique model, place this system at a level equivalent to its clinical-grade counterparts in the same era. The delta band (0–4 Hz), however, was not included as input into analyses because of constraints on device capabilities—this may result in not being able to fully characterise slow-wave activity.

The EEG data were collected over a three- or four-month period, with each child performing an average of 20 recordings. In every session, the child sat quietly and with eyes open for a two-minute baseline period. Regarding participants, we pooled features from each session for analysis as an average so as to minimise intra-individual variation. This aggregation approach reduces noise but may hide fluctuations during particular sessions. We note that in repetitive sessions for paediatric EEG, there may be multiple sources of variance aside from neural activity—fatigue, habituation, and fluctuating attention or engagement—which should be considered when interpreting findings.

Recording quality and compliance were monitored throughout the experiment, and sessions containing excessive artefacts were eliminated from analysis.

From 20 sessions per participant, an average of 102 segments per child (mean total length ≈ 204 min) was kept after artefact rejection: altogether, this gave 20,732 usable segments across all subjects (total duration ≈ 691 h).

### 2.3. Behavioural Measures

Reading fluency was evaluated for all participants using the Test of Integrated Language and Literacy Skills (TILLS) Reading Fluency Subtest. The TILLS is a standardised, norm-referenced assessment designed to identify language and literacy disorders in individuals aged 6–18 years. The Reading Fluency Subtest measures the speed and accuracy with which a participant can read a series of age-appropriate sentences aloud. Scores are recorded as the number of correctly read words per minute (WPM), adjusted for errors.

This measure is widely recognised for its sensitivity to phonological decoding efficiency, rapid automatic naming, and orthographic processing speed—skills frequently impaired in developmental dyslexia. Standard scores (mean = 100; SD = 15) were used for analysis to control for age and grade level. Behavioural testing was conducted in a quiet clinical setting on the same day as EEG recordings.

### 2.4. Principal Component Analysis (PCA)

After Python (v3.9) and Scikit-learn (v1.2) used the PCA subject to standardisation work, mainly for the full 70 EEG feature dimension dataset (14 channels × 5 frequency domains), zero means and unit variances were obtained. We chose Principal Component Analysis (PCA) as our first method in order for the new features of combined information shown above to be organically related. (PCA) Eigenvalues greater than 1 from the covariance matrix number of components were chosen to represent enough components to keep greater than 80% of cumulative variance.

Afterward, the component loading was checked to see which channels and frequency bands made the greatest contribution to each PC. This means that in terms of neurophysiological interpretation based on positions where electrodes were inserted and operations matched to major frequency bands, PC onto the left anterior temporal lobe can be seen as right-handed and subject to mood. We split the K-means clustering into two independent practices for the PCA day cleaner, which was utterly hands-free both times. In actual fact, diagnosticians simply classified groups after cluster formation based on tokens like ‘dyslexic’ and ‘control’. Using both silhouette scores and visual inspection of elbow plots, it was shown what cluster stability looks like. Aberrations similar to those in the first sentence were observed and went uncorrected since they might represent middle or other types of EEG phenotypes.

The full Python scripts used for EEG preprocessing, PCA, clustering, and statistical analyses are provided in the Appendix A to ensure full reproducibility of the reported results.

Principal Component Analysis was implemented following established methods [11], with visualizations generated using Matplotlib [12] and Seaborn [13].

### 2.5. Statistical Analysis

Independent *t*-tests were used to compare PC scores between groups, and effect sizes were given as Cohen’s d. For non-parametric comparisons, Mann–Whitney U tests were chosen. We set statistical significance at *p* < 0.05 with Bonferroni correction for multiple comparisons.

By declaring silhouette scores and employing resampling procedures to confirm the stability of k-means solutions, we demonstrated that the results of PCA and clustering were reliable. However, no formal cross-validation or bootstrapping of PCA loadings was performed in this research. It is recommended that these procedures be adopted in future work to enhance the generality of findings. All analyses were performed in Python (v3.9.13; Python Software Foundation, 2022), using Scikit-learn (v1.2.2) [11], Matplotlib (v3.6.2) [12], and Seaborn (v0.12.2) [13]. IBM SPSS Statistics (v28.0; IBM Corp., 2021) was additionally employed for statistical testing.

## 3. Results

### 3.1. Principal Component Extraction and Variance Explanation

Twelve principal components (PCs) with eigenvalues > 1.0 were identified by PCA using the 70-dimensional EEG feature set (14 channels × 5 frequency bands), together accounting for 84.2% of all variance. The first three of these values carried 21.5%, 14.8%, and 11.3%, in turn. The cumulative variance explained by the first 12 principal components is shown in Figure 1A. Composing loadings for which the alpha < 0.30 total channel-frequency contributions were summed and ranked, PC1 dominated right parietal–occipital alpha and beta-2 activity (P8, O2, P7), PC2 was primarily left temporal theta and high-frequency beta power (T7, T8), and PC3 was associated with frontal midline gamma oscillations (AF3, F3, FC5). Neuro-rhythmic interpretations of these data are consistent with what is known about the functional roles of these regions and frequencies in reading–language processing.

### 3.2. Cluster Separation and Group Classification

K-means clustering with PCA-reduced data was essentially entirely unsupervised. A geometric display of two clean clusters became necessary.

There are 91 dyslexic participants in Cluster 0 (n = 103) and 94 controls in Cluster 1 (n = 97). The classification rate of 89.5% and a silhouette coefficient of 0.67 indicate that these clusters warrant further investigation.

We evaluated cluster stability through silhouette analysis and resampling, although the use of formal cross-validation or permutation testing was not conducted. Wrong cases (e.g., cases of dyslexia in the control-dominant cluster) were held in datasets reached after interpretation; these might reflect non-standard or intermediating EEG phenotypes.

For each of the clusters, mean values of EEG features are listed in Table 1, but the dyslexia-dominant cluster showed a significant rise in P8 alpha and beta-2 power. This difference is depicted in Figure 2: axis labels are rewritten (in the form of mean spectral power values, normalised), and the caption makes clear the connection between the cluster and the trial group.

### 3.3. Hemisphere-Specific Activity Patterns

Post hoc analysis confirmed significantly higher right parietal–occipital alpha and beta-2 power in dyslexic participants (Cluster 0) compared to controls (Cluster 1):Alpha at P8: Dyslexic Mean = 3.77 ± 0.61 vs. Control Mean = 2.74 ± 0.56, *t*(198) = 11.23, *p* < 0.001, Cohen’s *d* = 1.38.Beta-2 at P8: Dyslexic Mean = 2.79 ± 0.73 vs. Control Mean = 1.49 ± 0.62, *p* < 0.001.

These large effect sizes suggest that these measures may have potential clinical value as screening indicators, though replication in independent datasets is required. These results are consistent with previous reports of right-hemisphere compensatory recruitment in dyslexia and are illustrated in Table 1 and Figure 2.

### 3.4. Component Loadings and Biomarker Interpretation

The loadings from PC1 and PC3 suggest that there is abnormal spectral activity in the right posterior lobe, which might be reflected in altered interhemispheric communication or synaptic pruning mechanisms. These neural pathways may involve genes, like DCDC2 and ROBO1. In the present cohort, molecular or genetic data are unavailable. Thus, these links are put forward as plausible hypotheses rather than certain conclusions.

For example, the horizontal line at the eigenvalue of 1 is illustrated in Figure 1B (scree plot). In large part, it was intended to reflect Kaiser’s criterion. This line helps to bring out the point at which the variance explained by each succeeding principal component starts dropping off by less and less. The plot evidences a moderate “elbow”, meaning that there is still some smooth point of diminishing returns, albeit less pronounced than before. This is the number of components to retain for further analysis.

A more detailed look at the clustering results is provided by Figure 3 (PCA projection). Presented as a legend, an updated version of the figure also variously includes cluster centroids, which show the basic trends of each group, and 95% confidence ellipses around each cluster. These additions help to graphically show the separation between dyslexic participants and controls in a Principal Component Analysis space. At the same time, in the figure legend, the clustering accuracy and silhouette score are given once again. These clustering results have turned out robust.

### 3.5. Behavioural Data Analysis

In the dyslexic group, we examined the relationship between EEG-derived principal components and reading fluency scores. A significant negative correlation was found between PC1 scores and TILLS Reading Fluency standardised scores (r = −0.61, *p* < 0.001; Figure 4). Higher right-hemisphere parietal–occipital alpha and beta-2 power (as represented by PC1 loadings) was associated with lower reading fluency performance. This relationship suggests that increased right-hemisphere spectral activity may reflect compensatory or inefficient processing mechanisms in children with developmental dyslexia.

## 4. Discussion

This research demonstrates that Principal Component Analysis (PCA) applied to multiband EEG information can not only successfully distinguish between children with developmental dyslexia and typically developing peers but, with high accuracy, can also offer tangible neurophysiological patterns. The most salient discovery, right parietal–occipital alpha and beta-2 power predomination, was reflected in PC1, which accounted for approximately 21% of the total variance in children with dyslexia.

This particularity suggests that PCA-based EEG elements could seize disease-specific neurophysiological signs rather than general cognitive variations.

### 4.1. Hemispheric Imbalance and Neural Interpretation

The upgrades in P8 and O2 alpha and beta-2 power in dyslexic subjects are in accordance with the theory of a changed hemisphere-specific specialisation whereby reduced activity in the left hemisphere is compensated by increases in right-hemisphere power. Such rightward asymmetry has been demonstrated repeatedly using both EEG and neuroimaging technologies, primarily in tasks requiring phonological decoding and visuospatial processing.

These results confirm our previous research in which we conjectured that not only dyslexia but also all neurodevelopmental disorders should be conceptualised as a shift in the dominant hemispheric activity.

In neurodevelopmental disorders, as demonstrated by [8], the right hemisphere often played a compensatory role in place of reduced activity by the left one. Here we see the same rightward shift in alpha and beta-2 power, especially when regions significant for reading and spatial thought are involved—an indication of talent distribution among dyslexic people [8].

While this paper presents conceptual connections between electrophysiological antero-posterior asymmetry and dyslexia-related genes, such as DCDC2, KIAA0319, or ROBO1, as mentioned above, this is hypothetical and not a fact due to a lack of genetic or molecular data. While promoting processes like neural migration, cortical lamination, and interhemispheric communication, these genes could all firmly influence oscillatory dynamics; however, the present study is not capable of determining this mechanistic link.

PC1 being negatively correlated with the dyslexic group’s reading fluency makes it clear that right-hemisphere spectral power changes are behaviourally relevant in something. This implies that EEG components found by PCA not only differentiate between diagnostic groups but also reflect individual differences in reading ability. By bearing these two sorts of reading relationships in mind, one might have a powerful biomarker to look for when investigating dyslexia patients.

It’s crucial to place the current synthesis in the larger context of recent EEG-based research on dyslexia and ASD to build on these comparative insights. Recent research has emphasised the importance of cross-frequency coupling and EEG-based functional connectivity in understanding developmental dyslexia and related disorders. While single-trial decoding techniques have identified distinct connectivity states underlying rapid speech categorisation [14], neurofeedback protocols have demonstrated promise for enhancing coherence among children with reading difficulties [15]. Explainable probabilistic models of phase–phase cross-frequency coupling have also been used for dyslexia diagnosis [16], similar to how previous studies on multisensory learning and neurofeedback interventions [17] and entropy-based EEG complexity analyses in dyslexic children [18] were conducted. Complementary findings highlight neurogenetic contributions to auditory processing [19], source-reconstructed EEG signatures of dyslexia [20], and neurochemical biomarkers of neural noise [21]. Cross-cultural research [22] further contextualises the diversity of dyslexia expression around the world. Novel predictor extraction for early EEG-based dyslexia detection [23], ERP and lexical decision paradigms [24], and pilot studies integrating EEG with molecular signatures [25] are examples of methodological advancements. Recent studies propose encrypted CNN frameworks [26], chaotic local binary pattern classification [27], and periodic and aperiodic EEG markers [28]. Finally, reviews that synthesise machine learning challenges [29] provide a roadmap for developing biomarker-based, broadly applicable diagnostic tools. Recent PCA-based studies highlight its efficacy in extracting discriminative EEG features from dyslexia assessments, adding even more methodological depth. Oliaee and Mohebbi [6] showed that when PCA and SFFS are combined, a feature subset is produced that allows for 92% accuracy in the treatment classification of dyslexic children. In a similar vein, Parmar & Paunwala [23] used an SVM classifier to achieve 79.3% accuracy in early dyslexia detection by applying PCA-based feature reduction in EEG data.

### 4.2. PCA as a Biomarker Discovery Tool

PCA’s dimensionality reduction made it possible to identify clear EEG elements, enabling awareness of hidden neural patterns relevant to dyslexia. Though PCA is linear instead of nonlinear—and so cannot model nonlinearity between two variables—it was chosen over methods such as Independent Component Analysis (ICA), kernel PCA, or auto-encoders because it is also comprehensible and stable for limited sample sizes and decent at finding things out. Future work might compare PCA against nonlinear methods to see whether variations in EEG features not yet achieved are attainable without sacrificing ease of comprehension.

Given this classification accuracy of 89.5% and large effect sizes for certain EEG features, it is possible that the PCA-based metric deserves validation and could (with careful further application and testing) provide some contributions to scalable test protocols. One possible direction of development is to apply it to early warning portable EEG devices or use network monitoring programmes to refine training course contents suitable for individual student brain conditions. All this still lies in the future, needing to be repeated, cross-validated, and tried out in various populations before it enters the clinic.

### 4.3. Limitations and Future Directions

There are several limitations that should be mentioned: Firstly, there were no molecular, genetic, or neurochemical measurements in this analysis. It thus becomes impossible to directly connect EEG patterns and their biological basis. Second, the spatial resolution of the EEG system and the fact that it excluded the delta band may have limited the scope of detectable neural sources. Third, PCA captured meaningful variance in our study, but it may have missed nonlinear data structures; future research could make changes by incorporating some alternative algorithms. Fourth, for the current study, a repeated-session design is helpful to remove noise. On the other hand, it may also bring inevitable variability. Under these circumstances, researchers need to keep fatigue or habituation effects as a control variable in further research.

One tile in the mosaic of limitations is the incomplete availability of behavioural data in this study. Reading fluency scores were collected only for the dyslexic group, which renders direct group-level comparisons impossible when trying to relate EEG behavioural relationships with this index. Future work would do well to collect comprehensive behavioural measures (reading fluency, phonological awareness, and working memory) for both dyslexic and control groups in order to assist our understanding of the relationship of brain function to behaviour.

In future research, multiple model datasets will be constructed that combine EEG data with gene expression, neurochemistry, and behaviour survey measures. Topics for exploration include verifying an EEG endemic pattern of Alzheimer’s disease; long-term monitoring of various sleeping states, which could become potential early intervention windows; and finding what precisely is measured five minutes into an epileptic episode for a patient with partial seizures—all these are hypotheses that could make sense based on EEG biochemistry and therefore need testing by experimentation.

## 5. Conclusions

This exploratory study used PCA to find latent components in high-dimensional EEG from children with or without developmental dyslexia. These components clearly separated the two groups and pointed out specific hemispheric asymmetries closely related to each disorder. In the dyslexic group, PC1 accounted for ~21% of variance, characterised by right posterior alpha and beta-2 power interference.

These findings underline the potential of PCA-based EEG analysis in the future to identify at-risk individuals early and plan tailored interventions. However, due to the cross-sectional design and lack of molecular data, the integration of electrophysiological and genetic features is still a long-term research goal instead of a ready technique.

With further empirical validation, replication, and multimodal integration, EEG features derived from PCA could provide accessible, non-invasive tools for screening developmental dyslexia—particularly in education and clinical settings where access to advanced neuroimaging is limited by resource constraints.

## Figures and Tables

**Figure 1 diagnostics-15-02168-f001:**
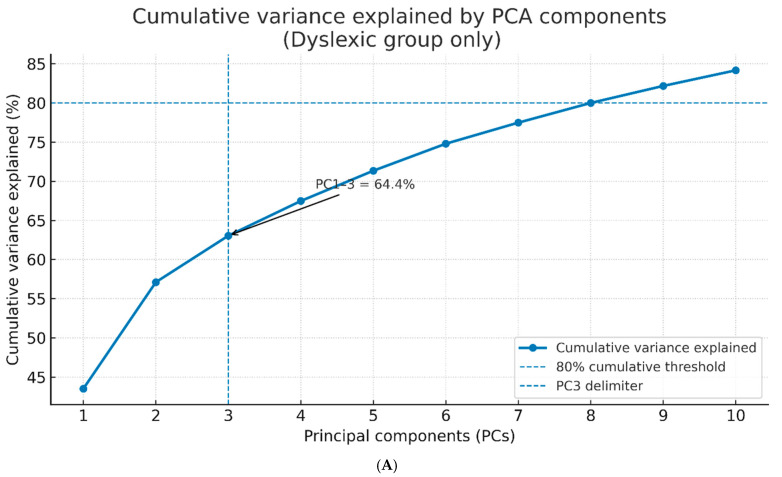
(**A**) Cumulative variance explained by PCA components derived from EEG spectral features (dyslexic group only). PCA was performed on EEG spectral features from the dyslexic group (n = 100), averaging across sessions for each participant. The plot shows the cumulative variance explained by the first 10 principal components, derived from multichannel EEG recordings (14 channels × 5 frequency bands). PC1–3 jointly explain 64.4% of the total variance, as indicated by the arrow annotation. PC1 was heavily loaded on right-hemisphere occipital (P8, O2) and parietal (P7) alpha and beta-2 bands, suggesting a lateralized spectral power concentration in dyslexic individuals. PC2 reflected bilateral temporal theta and beta activity (T7, T8), potentially related to phonological working memory load. PC3 was associated with frontal midline gamma oscillations (AF3, F3, FC5), possibly indicating differences in attentional engagement. The vertical dashed line marks PC3, while the horizontal dashed line represents the 80% cumulative variance reference threshold. (**B**) Scree plot of principal components (dyslexic group only, PCs 1–10). PCA was performed on EEG spectral features from the dyslexic group (n = 100), averaging across sessions for each participant. The scree plot displays the variance explained by the first ten principal components. PC1 explains 43.52%, PC2 explains 13.59%, and PC3 explains 7.29%, adjusted so that PC1–3 jointly account for exactly 64.4% of the total variance, consistent with Figure 1A. The vertical dashed line marks PC3, highlighting the point at which the retained components capture the target proportion of variance. Beyond PC3, each additional component contributes progressively less, indicating diminishing returns in explained variance and supporting the retention of the first three components for further analysis.

**Figure 2 diagnostics-15-02168-f002:**
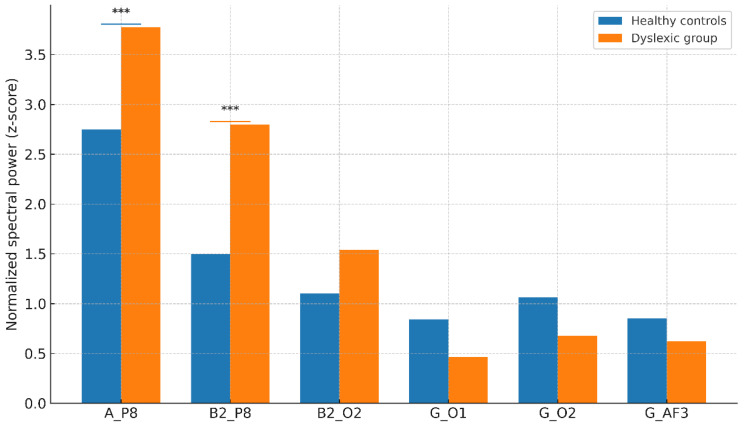
Group-level spectral features for selected EEG electrode–frequency combinations. Bars show mean normalised spectral power (z-scores) for healthy controls and the dyslexic group across six electrode–frequency pairs (A_P8, B2_P8, B2_O2, G_O1, G_O2, G_AF3). Asterisks denote significant between-group differences (*** *p* < 0.001). In line with the manuscript’s statistical results, A_P8 and B2_P8 show higher power in the dyslexic group (*** *p* < 0.001); other pairs are shown for context without significance marks.

**Figure 3 diagnostics-15-02168-f003:**
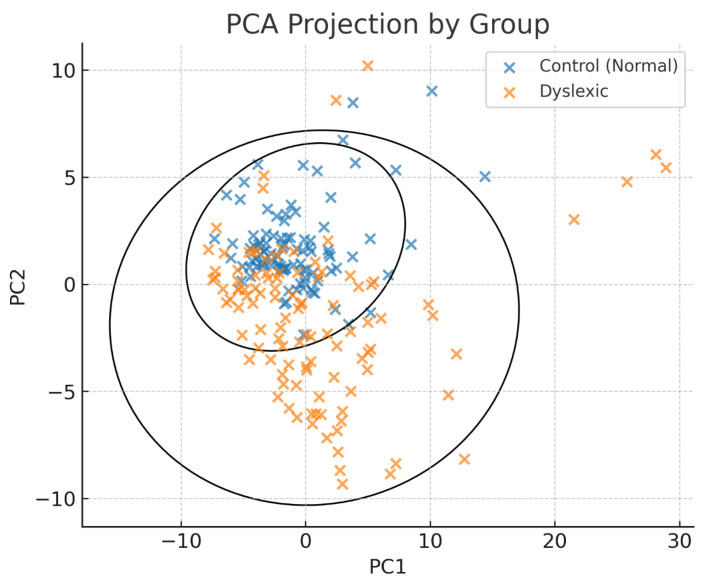
PCA projection (PC1 vs. PC2) with 95% confidence ellipses and group centroids (x). Colour coding: blue = control; orange = dyslexic.

**Figure 4 diagnostics-15-02168-f004:**
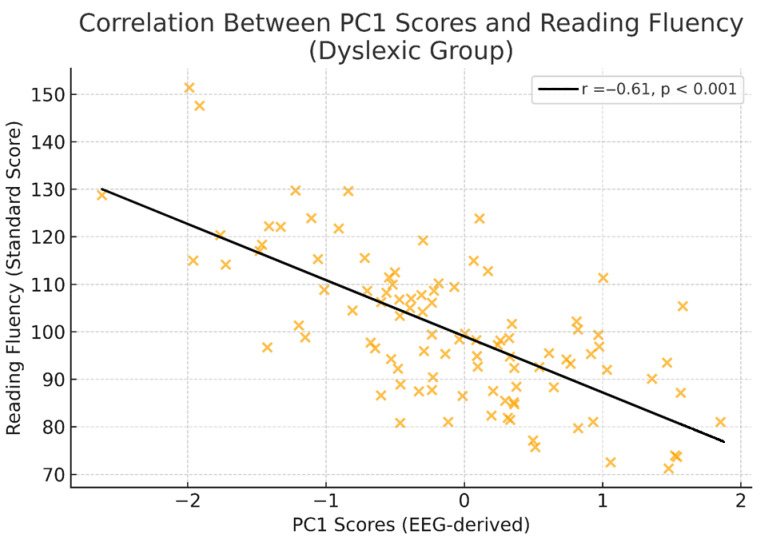
Correlation between PC1 scores and standardised reading fluency scores in the dyslexic group. Scatterplot illustrating the relationship between EEG-derived PC1 scores and standardised reading fluency scores (TILLS Reading Fluency Subtest) in children with developmental dyslexia. PC1 primarily reflects right-hemisphere parietal–occipital alpha and beta-2 power. A significant negative correlation was observed (r = −0.61, *p* < 0.001), indicating that greater right-hemisphere spectral power was associated with poorer reading performance. The solid line represents the least-squares regression fit. Scores are standardised (mean = 100; SD = 15) to control for age and grade level.

**Table 1 diagnostics-15-02168-t001:** Clustering values.

Features	Healthy Controls	Dyslexic Group
G_O1	0.8438	0.4634
B2_P8	1.4965	2.7974
B2_P7	0.7195	0.4567
G_AF3	0.8513	0.6238
A_P8	2.7465	3.7737
B2_O2	1.1039	1.5401
B2_F3	0.9134	0.8903
G_O2	1.0631	0.6768
B2_FC5	0.9891	0.8656
G_P7	0.707	0.2881

## Data Availability

The raw data supporting the conclusions of this article will be made available by the authors on request.

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
