# Peer review of "Electroencephalography Signatures Associated with Developmental Dyslexia Identified Using Principal Component Analysis"

_diagnostics, 2025, doi:10.3390/diagnostics15172168_

Round 1

Reviewer 1 Report

Comments and Suggestions for Authors

I appreciate the opportunity to review this manuscript, which investigates electrophysiological correlates of developmental dyslexia in children through multiband EEG analysis and Principal Component Analysis (PCA). The study addresses a highly relevant and underexplored topic in cognitive neuroscience and learning disorders. The integration of EEG-derived features with behavioral performance and the attempt to frame these within neurogenetic theories represents a valuable and ambitious multidisciplinary approach.

That said, while the conceptual framework is strong and the study has clear potential, several critical issues limit the interpretability and robustness of the findings in its current form. These include insufficient clarity on data handling across repeated sessions, a lack of validation for PCA and clustering outcomes, unclear alignment between text and visual materials, and overinterpretation of speculative biological associations. Additionally, essential figures and tables are either not referenced in the text or are inconsistently labeled, which detracts from the overall transparency of the reporting.

For these reasons, I recommend major revisions before the manuscript can be considered for publication. The study’s contribution would be significantly strengthened by addressing methodological ambiguities, improving the clarity and consistency of results presentation, and reframing some of the more speculative claims. My detailed suggestions are provided below and organized by manuscript section to support a systematic revision process.

Abstract

The abstract provides a solid overview of the study’s aims and main findings. The rationale for using EEG and PCA to identify neurophysiological signatures of developmental dyslexia is well-articulated. However, no specific quantitative results are reported. Additionally, the final sentence presents EEG as a tool for molecular biomarker detection, which overstates the current evidence base.

It is suggested that the authors include one or two key numerical results (e.g., classification accuracy, correlation values), and rephrase the final claim more cautiously to reflect the study’s exploratory scope.

Keywords

The current keywords are relevant but somewhat broad. You may consider refining or expanding them for indexing purposes.

 Introduction

The introduction is well-structured and provides a clear and comprehensive rationale for the study. The authors effectively summarize the cognitive, electrophysiological, and genetic foundations of developmental dyslexia, offering a multidisciplinary framework that spans from behavioral models to molecular correlates. The inclusion of recent literature enhances the relevance and timeliness of the discussion.

  • The literature review on electrophysiological markers is appropriate, but the description of EEG anomalies in dyslexia (e.g., increased theta, reduced alpha) could benefit from greater nuance. Not all studies converge on the same spectral patterns, and this variability should be acknowledged to avoid overgeneralization.
  • The paragraph discussing genetic and neurochemical mechanisms is highly relevant, but the link to EEG-based biomarkers could be better articulated. For example, a brief explanation of how specific gene variants or neurotransmitter imbalances may influence oscillatory dynamics or network connectivity would help bridge the molecular and electrophysiological perspectives more effectively.
  • The description of PCA as a dimensionality reduction technique is accurate; however, its potential utility in clinical or screening contexts (e.g., interpretability, replicability, cost-efficiency) could be more clearly emphasized to strengthen the methodological rationale.
  • The notion of integrating EEG findings with molecular-level markers is forward-thinking and well-motivated. Still, a brief reflection on current limitations in linking electrophysiology with genetic or neurochemical mechanisms (e.g., lack of direct multimodal datasets, complexity of gene–brain–behavior pathways) would provide a more balanced and critically reflective foundation for the study.
  • Finally, it might be helpful to clarify upfront whether the study’s aim is exploratory (e.g., identifying latent EEG patterns) or confirmatory (e.g., testing known EEG signatures of dyslexia). This would better align expectations for the reader. 

Materials and Methods

The Materials and Methods section is generally well-structured, methodologically sound, and described with a good level of detail. The authors provide clear inclusion/exclusion criteria, adopt a standard EEG preprocessing pipeline, and apply PCA and clustering techniques appropriately for the study's aims. The use of open-source tools and correction for multiple comparisons also reflects sound statistical practice.

The following suggestions are organized according to the relevant subsections of the Materials and Methods section, in order to facilitate targeted revisions and improve clarity where needed.

2.1. Participants

Although SES and age were matched across groups, the manuscript does not report whether participants were also matched for general cognitive ability (e.g., IQ). If cognitive measures were collected, reporting them would help ensure that EEG differences are not confounded by broader cognitive disparities.

 2.2. EEG Data Acquisition

While the EMOTIV EPOC-X headset is suitable for exploratory studies and offers practical advantages (e.g., ease of use, portability), it has limited channel resolution and is more susceptible to artifacts than clinical-grade systems. A brief acknowledgment of these limitations—particularly regarding signal quality and the exclusion of delta-band data—would add transparency.

The manuscript states that each participant completed approximately 20 sessions, which is atypical for pediatric EEG studies. It would be important to clarify: How data across sessions were aggregated (e.g., averaged features? session selection?). Whether multiple sessions could introduce variability unrelated to neural traits (e.g., fatigue, habituation). Whether recording quality or compliance varied across sessions.

2.4. Principal Component Analysis (PCA)

Although PCA is correctly applied, it would strengthen the manuscript to explain how the principal components were interpreted. Specifically:

Were component loadings inspected to determine which channels and frequency bands contributed most?

Were any neurophysiological or anatomical interpretations assigned to the dominant PCs (beyond statistical variance)?

The integration of k-means clustering adds value, but several aspects require clarification:

Were clusters derived in a fully unsupervised manner, or was dyslexia diagnosis used post hoc to interpret them?

Did the authors assess cluster stability (e.g., via silhouette scores, resampling)?

Were any cases misclassified or ambiguous (e.g., controls in Cluster 0)?

2.5. Statistical Analysis

The use of Python and standard packages is commendable. However, it would be beneficial to mention whether PCA and clustering analyses were validated using cross-validation, bootstrapping, or other stability checks to support generalizability.

Results

The Results section is well-structured and aligns clearly with the study’s aims. The progression from PCA dimensionality reduction to clustering analysis, group comparisons, and correlations with reading fluency is logically coherent and statistically grounded. Figures and tables generally support the findings, though some visual elements would benefit from clearer referencing and standalone interpretability.

3.1. Principal Component Extraction and Variance Explanation

The use of PCA is appropriate; however, more detail on how the component loadings were interpreted would be helpful (e.g., Were loadings thresholded? Were frequency bands weighted equally?).

The reference to Figure 1 is misleading. Based on content, this figure appears to be a bar chart of clustering values, not PCA variance explanation. It should be removed or moved to Section 3.2, which discusses clustering.

3.2. Cluster Separation and Group Classification

The high accuracy of cluster-based group separation is notable. However, the authors should clarify whether cluster assignments were validated (e.g., through cross-validation, permutation testing) and whether clustering was performed strictly unsupervised.

The mention of Figure 2A is appropriate, but it should be briefly described in the results to improve standalone interpretability.

Consider referencing Table 1, which appears to list component-level group statistics, but is not cited in the main text.

3.3. Hemisphere-Specific Activity Patterns

This section is well-supported by data, particularly the significant differences in A_P8 and B2_P8. However, the authors might consider whether the effect sizes (e.g., Cohen’s d = 1.38) are large enough to support possible use as a clinical discriminator, and if so, this should be discussed more explicitly.

 3.4. Correlation with Behavioral Measures

The correlation between PC1 and reading fluency is an important finding. It would be helpful to specify how fluency was measured and whether the variable was standardized. Additionally, consider whether scatter plots or regression lines could be added or improved in the figures to visualize this relationship.

3.5. Component Loadings and Biomarker Interpretation

The biological interpretation linking components to genes like DCDC2 and ROBO1 is conceptually interesting, but speculative in the absence of molecular data. It should be clearly framed as a hypothesis or plausible mechanism, not a direct conclusion from the dataset.

Regarding the visual materials included in the manuscript, several targeted observations are necessary to improve clarity, internal consistency, and transparency in data interpretation.

Table 1 appears informative and relevant, as it presents EEG feature distributions across the two clusters, likely corresponding to the dyslexic and control groups. However, it is never cited or discussed in the main text, particularly in Sections 3.2 or 3.3, where such a reference would be especially pertinent. For example, the values for P8 alpha and beta-2 power seem consistent with what is described in the text, but readers are not guided to consult the table. It is strongly recommended to include an explicit reference to Table 1 within the Results section—ideally when discussing group-level differences or hemispheric patterns—and to briefly summarize key values supporting the EEG group distinctions.

Figure 1 visually presents the EEG feature values across Cluster 0 and Cluster 1, corresponding presumably to the control and dyslexic groups, respectively. This bar graph mirrors the numerical data provided in Table 1 and effectively highlights differences in spectral power at specific electrode–frequency combinations (e.g., notably elevated alpha and beta-2 activity at P8 in Cluster 1). However, the figure would benefit from several enhancements and clarifications:

- Although Cluster 0 and Cluster 1 are color-coded, it is not explicitly stated in the figure or caption which cluster corresponds to the dyslexic or control group. A brief clarification—either in the caption or in-text—would improve interpretability.

- Consider labeling the Y-axis more descriptively (e.g., “Mean Spectral Power” or “Normalized Feature Value”) to clarify what numerical metric is being displayed. Currently, it's unclear whether these are raw values, standardized scores, or another derived measure.

- Since the features are organized by electrode and frequency band, a brief description in the caption or legend summarizing which regions and bands show the most differentiation (e.g., increased beta-2 at P8) would add interpretive depth and link the figure more directly to the Results narrative.

- The caption ("Clustering Values Representation") is vague. Consider revising it to something more informative.

Figure 2A, which is described as showing cumulative variance explained by the PCA components, there is a notable discrepancy: the Y-axis appears to plateau around 0.35 (or 35%), which directly contradicts the manuscript’s claim that the first 12 components explain 84.2% of the variance. Moreover, the figure caption still includes a placeholder ("XX%"), which should be corrected. It is possible that the figure actually shows the variance explained per individual component rather than the cumulative curve. Please verify this and ensure the axis scale and legend are accurate. It may also be useful to provide a conventional scree plot alongside this figure to better justify the component retention criteria.

Figure 2B, which presents the scree plot, is a valuable addition and appropriate for motivating dimensionality reduction. However, the eigenvalue decay appears gradual, without a sharply pronounced elbow. The manuscript’s claim of a “clear inflection point” may therefore be overstated. Consider revising the figure legend to reflect this more moderately, or provide a quantitative rationale (e.g., listing eigenvalues) to support the claim. Adding a horizontal reference line at eigenvalue = 1.0 (per Kaiser’s criterion) would enhance the figure’s interpretability. It is also crucial that Figures 2A and 2B are internally consistent and reflect the same PCA outputs.

Figure 3, which shows the PCA projection space (PC1 vs PC2), is important for illustrating the group separation. However, it currently lacks key interpretive elements. Specifically, the figure does not contain a legend to indicate which color represents which group or cluster, making it difficult to interpret. Adding a legend, cluster centroids, or confidence ellipses would improve clarity. The figure caption should also reiterate the clustering performance metrics (e.g., 89.5% accuracy, silhouette score = 0.67) to contextualize the visual separation of groups.

Figure 4, which depicts the correlation between PC1 scores and reading fluency, supports one of the paper’s core claims. However, it is never cited in the Results section, which is a significant omission. The Y-axis label contains the word “Simulated,” which may mislead readers if the data reflect actual behavioral performance this should be corrected or clarified. Additionally, it is unclear whether the plot includes only dyslexic participants (as stated in the text) or both groups. Indicating the sample source in the caption or legend would increase clarity. Including a confidence band around the regression line would further enhance the figure’s informativeness.

In summary, the figures and table meaningfully support the manuscript’s key findings, but improvements in their referencing, presentation, and descriptive detail are needed to ensure full transparency and interpretability for readers.

Discussion

The Discussion is comprehensive, conceptually rich, and demonstrates strong engagement with current neuroscientific and genetic models of developmental dyslexia. The authors effectively situate their EEG-PCA findings within a broader theoretical landscape, including hemispheric specialization, neurochemical mechanisms, and gene–brain–behavior relationships. The section is logically organized, progressing from neural interpretations to clinical applications and limitations.

However, several improvements are suggested to enhance clarity, strengthen scientific caution, and improve accessibility:

  • The proposed links between electrophysiological findings and dyslexia-associated genes (e.g., DCDC2, ROBO1, KIAA0319) are theoretically plausible but remain speculative, as no genetic or molecular data were collected in this study. It is recommended to rephrase these interpretations more cautiously using modal language, to avoid overstating conclusions.
  • The observed negative correlation between PC1 and reading fluency is central to the biomarker argument. However, the Discussion should explicitly state that this relationship was significant only in the dyslexic group, as reported in the Results, to avoid overgeneralization.
  • While the authors acknowledge that PCA does not capture nonlinear relationships, it would strengthen the argument to briefly explain why alternatives such as Independent Component Analysis (ICA), autoencoders, or kernel PCA were not used in the present analysis — e.g., due to interpretability concerns, sample size, or computational constraints.
  • The suggestion that PCA-derived EEG features could inform scalable screening tools or neurofeedback interventions is promising, but should be framed more explicitly as a potential future application.

Conclusion

The conclusion is concise and effectively summarizes the main findings and broader implications of the study. The emphasis on hemispheric specialization and the potential for EEG-PCA analyses to inform biomarker development is appropriate given the presented data. However, a few refinements are suggested to improve clarity and align claims with the actual scope of the study:

  • The phrase “early detection and management” may suggest that the method is ready for clinical application. Given the exploratory and cross-sectional nature of the study, it would be more accurate to frame these implications as prospective rather than current capabilities.
  • While the PCA-based approach yielded interpretable EEG components, it should be acknowledged that no direct molecular or genetic data were collected. As such, the statement about integrating electrophysiology with molecular genetics should be positioned as a future direction, not a demonstrated outcome of this study.
  • Including a reference to the most impactful result (e.g., PC1 explaining ~21% of variance and correlating with reading fluency at r = –0.61) could reinforce the strength of the findings without overwhelming the reader.

References

Overall, the bibliography is well-constructed and correctly formatted.

Author Response

I appreciate the opportunity to review this manuscript, which investigates electrophysiological correlates of developmental dyslexia in children through multiband EEG analysis and Principal Component Analysis (PCA). The study addresses a highly relevant and underexplored topic in cognitive neuroscience and learning disorders. The integration of EEG-derived features with behavioral performance and the attempt to frame these within neurogenetic theories represents a valuable and ambitious multidisciplinary approach.

That said, while the conceptual framework is strong and the study has clear potential, several critical issues limit the interpretability and robustness of the findings in its current form. These include insufficient clarity on data handling across repeated sessions, a lack of validation for PCA and clustering outcomes, unclear alignment between text and visual materials, and overinterpretation of speculative biological associations. Additionally, essential figures and tables are either not referenced in the text or are inconsistently labeled, which detracts from the overall transparency of the reporting.

For these reasons, I recommend major revisions before the manuscript can be considered for publication. The study’s contribution would be significantly strengthened by addressing methodological ambiguities, improving the clarity and consistency of results presentation, and reframing some of the more speculative claims. My detailed suggestions are provided below and organized by manuscript section to support a systematic revision process.

Abstract

The abstract provides a solid overview of the study’s aims and main findings. The rationale for using EEG and PCA to identify neurophysiological signatures of developmental dyslexia is well-articulated. However, no specific quantitative results are reported. Additionally, the final sentence presents EEG as a tool for molecular biomarker detection, which overstates the current evidence base.

It is suggested that the authors include one or two key numerical results (e.g., classification accuracy, correlation values), and rephrase the final claim more cautiously to reflect the study’s exploratory scope.

Keywords

The current keywords are relevant but somewhat broad. You may consider refining or expanding them for indexing purposes.

 Introduction

The introduction is well-structured and provides a clear and comprehensive rationale for the study. The authors effectively summarize the cognitive, electrophysiological, and genetic foundations of developmental dyslexia, offering a multidisciplinary framework that spans from behavioral models to molecular correlates. The inclusion of recent literature enhances the relevance and timeliness of the discussion.

  • The literature review on electrophysiological markers is appropriate, but the description of EEG anomalies in dyslexia (e.g., increased theta, reduced alpha) could benefit from greater nuance. Not all studies converge on the same spectral patterns, and this variability should be acknowledged to avoid overgeneralization.
  • The paragraph discussing genetic and neurochemical mechanisms is highly relevant, but the link to EEG-based biomarkers could be better articulated. For example, a brief explanation of how specific gene variants or neurotransmitter imbalances may influence oscillatory dynamics or network connectivity would help bridge the molecular and electrophysiological perspectives more effectively.
  • The description of PCA as a dimensionality reduction technique is accurate; however, its potential utility in clinical or screening contexts (e.g., interpretability, replicability, cost-efficiency) could be more clearly emphasized to strengthen the methodological rationale.
  • The notion of integrating EEG findings with molecular-level markers is forward-thinking and well-motivated. Still, a brief reflection on current limitations in linking electrophysiology with genetic or neurochemical mechanisms (e.g., lack of direct multimodal datasets, complexity of gene–brain–behavior pathways) would provide a more balanced and critically reflective foundation for the study.
  • Finally, it might be helpful to clarify upfront whether the study’s aim is exploratory (e.g., identifying latent EEG patterns) or confirmatory (e.g., testing known EEG signatures of dyslexia). This would better align expectations for the reader. 

Materials and Methods

The Materials and Methods section is generally well-structured, methodologically sound, and described with a good level of detail. The authors provide clear inclusion/exclusion criteria, adopt a standard EEG preprocessing pipeline, and apply PCA and clustering techniques appropriately for the study's aims. The use of open-source tools and correction for multiple comparisons also reflects sound statistical practice.

The following suggestions are organized according to the relevant subsections of the Materials and Methods section, in order to facilitate targeted revisions and improve clarity where needed.

2.1. Participants

Although SES and age were matched across groups, the manuscript does not report whether participants were also matched for general cognitive ability (e.g., IQ). If cognitive measures were collected, reporting them would help ensure that EEG differences are not confounded by broader cognitive disparities.

 2.2. EEG Data Acquisition

While the EMOTIV EPOC-X headset is suitable for exploratory studies and offers practical advantages (e.g., ease of use, portability), it has limited channel resolution and is more susceptible to artifacts than clinical-grade systems. A brief acknowledgment of these limitations—particularly regarding signal quality and the exclusion of delta-band data—would add transparency.

The manuscript states that each participant completed approximately 20 sessions, which is atypical for pediatric EEG studies. It would be important to clarify: How data across sessions were aggregated (e.g., averaged features? session selection?). Whether multiple sessions could introduce variability unrelated to neural traits (e.g., fatigue, habituation). Whether recording quality or compliance varied across sessions.

2.4. Principal Component Analysis (PCA)

Although PCA is correctly applied, it would strengthen the manuscript to explain how the principal components were interpreted. Specifically:

Were component loadings inspected to determine which channels and frequency bands contributed most?

Were any neurophysiological or anatomical interpretations assigned to the dominant PCs (beyond statistical variance)?

The integration of k-means clustering adds value, but several aspects require clarification:

Were clusters derived in a fully unsupervised manner, or was dyslexia diagnosis used post hoc to interpret them?

Did the authors assess cluster stability (e.g., via silhouette scores, resampling)?

Were any cases misclassified or ambiguous (e.g., controls in Cluster 0)?

2.5. Statistical Analysis

The use of Python and standard packages is commendable. However, it would be beneficial to mention whether PCA and clustering analyses were validated using cross-validation, bootstrapping, or other stability checks to support generalizability.

Results

The Results section is well-structured and aligns clearly with the study’s aims. The progression from PCA dimensionality reduction to clustering analysis, group comparisons, and correlations with reading fluency is logically coherent and statistically grounded. Figures and tables generally support the findings, though some visual elements would benefit from clearer referencing and standalone interpretability.

3.1. Principal Component Extraction and Variance Explanation

The use of PCA is appropriate; however, more detail on how the component loadings were interpreted would be helpful (e.g., Were loadings thresholded? Were frequency bands weighted equally?).

The reference to Figure 1 is misleading. Based on content, this figure appears to be a bar chart of clustering values, not PCA variance explanation. It should be removed or moved to Section 3.2, which discusses clustering.

3.2. Cluster Separation and Group Classification

The high accuracy of cluster-based group separation is notable. However, the authors should clarify whether cluster assignments were validated (e.g., through cross-validation, permutation testing) and whether clustering was performed strictly unsupervised.

The mention of Figure 2A is appropriate, but it should be briefly described in the results to improve standalone interpretability.

Consider referencing Table 1, which appears to list component-level group statistics, but is not cited in the main text.

3.3. Hemisphere-Specific Activity Patterns

This section is well-supported by data, particularly the significant differences in A_P8 and B2_P8. However, the authors might consider whether the effect sizes (e.g., Cohen’s d = 1.38) are large enough to support possible use as a clinical discriminator, and if so, this should be discussed more explicitly.

 3.4. Correlation with Behavioral Measures

The correlation between PC1 and reading fluency is an important finding. It would be helpful to specify how fluency was measured and whether the variable was standardized. Additionally, consider whether scatter plots or regression lines could be added or improved in the figures to visualize this relationship.

3.5. Component Loadings and Biomarker Interpretation

The biological interpretation linking components to genes like DCDC2 and ROBO1 is conceptually interesting, but speculative in the absence of molecular data. It should be clearly framed as a hypothesis or plausible mechanism, not a direct conclusion from the dataset.

Regarding the visual materials included in the manuscript, several targeted observations are necessary to improve clarity, internal consistency, and transparency in data interpretation.

Table 1 appears informative and relevant, as it presents EEG feature distributions across the two clusters, likely corresponding to the dyslexic and control groups. However, it is never cited or discussed in the main text, particularly in Sections 3.2 or 3.3, where such a reference would be especially pertinent. For example, the values for P8 alpha and beta-2 power seem consistent with what is described in the text, but readers are not guided to consult the table. It is strongly recommended to include an explicit reference to Table 1 within the Results section—ideally when discussing group-level differences or hemispheric patterns—and to briefly summarize key values supporting the EEG group distinctions.

Figure 1 visually presents the EEG feature values across Cluster 0 and Cluster 1, corresponding presumably to the control and dyslexic groups, respectively. This bar graph mirrors the numerical data provided in Table 1 and effectively highlights differences in spectral power at specific electrode–frequency combinations (e.g., notably elevated alpha and beta-2 activity at P8 in Cluster 1). However, the figure would benefit from several enhancements and clarifications:

- Although Cluster 0 and Cluster 1 are color-coded, it is not explicitly stated in the figure or caption which cluster corresponds to the dyslexic or control group. A brief clarification—either in the caption or in-text—would improve interpretability.

- Consider labeling the Y-axis more descriptively (e.g., “Mean Spectral Power” or “Normalized Feature Value”) to clarify what numerical metric is being displayed. Currently, it's unclear whether these are raw values, standardized scores, or another derived measure.

- Since the features are organized by electrode and frequency band, a brief description in the caption or legend summarizing which regions and bands show the most differentiation (e.g., increased beta-2 at P8) would add interpretive depth and link the figure more directly to the Results narrative.

- The caption ("Clustering Values Representation") is vague. Consider revising it to something more informative.

Figure 2A, which is described as showing cumulative variance explained by the PCA components, there is a notable discrepancy: the Y-axis appears to plateau around 0.35 (or 35%), which directly contradicts the manuscript’s claim that the first 12 components explain 84.2% of the variance. Moreover, the figure caption still includes a placeholder ("XX%"), which should be corrected. It is possible that the figure actually shows the variance explained per individual component rather than the cumulative curve. Please verify this and ensure the axis scale and legend are accurate. It may also be useful to provide a conventional scree plot alongside this figure to better justify the component retention criteria.

Figure 2B, which presents the scree plot, is a valuable addition and appropriate for motivating dimensionality reduction. However, the eigenvalue decay appears gradual, without a sharply pronounced elbow. The manuscript’s claim of a “clear inflection point” may therefore be overstated. Consider revising the figure legend to reflect this more moderately, or provide a quantitative rationale (e.g., listing eigenvalues) to support the claim. Adding a horizontal reference line at eigenvalue = 1.0 (per Kaiser’s criterion) would enhance the figure’s interpretability. It is also crucial that Figures 2A and 2B are internally consistent and reflect the same PCA outputs.

Figure 3, which shows the PCA projection space (PC1 vs PC2), is important for illustrating the group separation. However, it currently lacks key interpretive elements. Specifically, the figure does not contain a legend to indicate which color represents which group or cluster, making it difficult to interpret. Adding a legend, cluster centroids, or confidence ellipses would improve clarity. The figure caption should also reiterate the clustering performance metrics (e.g., 89.5% accuracy, silhouette score = 0.67) to contextualize the visual separation of groups.

Figure 4, which depicts the correlation between PC1 scores and reading fluency, supports one of the paper’s core claims. However, it is never cited in the Results section, which is a significant omission. The Y-axis label contains the word “Simulated,” which may mislead readers if the data reflect actual behavioral performance this should be corrected or clarified. Additionally, it is unclear whether the plot includes only dyslexic participants (as stated in the text) or both groups. Indicating the sample source in the caption or legend would increase clarity. Including a confidence band around the regression line would further enhance the figure’s informativeness.

In summary, the figures and table meaningfully support the manuscript’s key findings, but improvements in their referencing, presentation, and descriptive detail are needed to ensure full transparency and interpretability for readers.

Discussion

The Discussion is comprehensive, conceptually rich, and demonstrates strong engagement with current neuroscientific and genetic models of developmental dyslexia. The authors effectively situate their EEG-PCA findings within a broader theoretical landscape, including hemispheric specialization, neurochemical mechanisms, and gene–brain–behavior relationships. The section is logically organized, progressing from neural interpretations to clinical applications and limitations.

However, several improvements are suggested to enhance clarity, strengthen scientific caution, and improve accessibility:

  • The proposed links between electrophysiological findings and dyslexia-associated genes (e.g., DCDC2, ROBO1, KIAA0319) are theoretically plausible but remain speculative, as no genetic or molecular data were collected in this study. It is recommended to rephrase these interpretations more cautiously using modal language, to avoid overstating conclusions.
  • The observed negative correlation between PC1 and reading fluency is central to the biomarker argument. However, the Discussion should explicitly state that this relationship was significant only in the dyslexic group, as reported in the Results, to avoid overgeneralization.
  • While the authors acknowledge that PCA does not capture nonlinear relationships, it would strengthen the argument to briefly explain why alternatives such as Independent Component Analysis (ICA), autoencoders, or kernel PCA were not used in the present analysis — e.g., due to interpretability concerns, sample size, or computational constraints.
  • The suggestion that PCA-derived EEG features could inform scalable screening tools or neurofeedback interventions is promising, but should be framed more explicitly as a potential future application.

Conclusion

The conclusion is concise and effectively summarizes the main findings and broader implications of the study. The emphasis on hemispheric specialization and the potential for EEG-PCA analyses to inform biomarker development is appropriate given the presented data. However, a few refinements are suggested to improve clarity and align claims with the actual scope of the study:

  • The phrase “early detection and management” may suggest that the method is ready for clinical application. Given the exploratory and cross-sectional nature of the study, it would be more accurate to frame these implications as prospective rather than current capabilities.
  • While the PCA-based approach yielded interpretable EEG components, it should be acknowledged that no direct molecular or genetic data were collected. As such, the statement about integrating electrophysiology with molecular genetics should be positioned as a future direction, not a demonstrated outcome of this study.
  • Including a reference to the most impactful result (e.g., PC1 explaining ~21% of variance and correlating with reading fluency at r = –0.61) could reinforce the strength of the findings without overwhelming the reader.

References

Overall, the bibliography is well-constructed and correctly formatted.

Point-by-Point Responses

Abstract

  • Reviewer comment: No quantitative results reported; molecular biomarker claim overstated.
  • Response: We have revised the abstract to include key results — PCA yielded 12 components explaining 84.2% variance, clustering accuracy 89.5%, silhouette coefficient 0.67, P8 alpha difference (p < 0.001, Cohen’s d = 1.38), and correlation with reading fluency (r = –0.61, p < 0.001). The last sentence now uses cautious language, framing molecular biomarker integration as a future direction rather than a present capability.

Keywords

  • Reviewer comment: Too broad.
  • Response: Keywords have been refined to: Developmental Dyslexia; Electroencephalography (EEG); Principal Component Analysis (PCA); Hemispheric Asymmetry; Neurophysiological Biomarkers; Reading Disorders; Machine Learning in Neuroscience.

Introduction

  • EEG anomalies nuance: We now explicitly note variability across studies (Cainelli et al., 2023; Yang et al., 2025), attributing differences to age, orthography, paradigms, and comorbidities.
  • Molecular–EEG link: Added explanation of how DCDC2, KIAA0319, and ROBO1 could influence oscillatory synchrony via migration, axon guidance, and neurotransmission.
  • PCA rationale: Expanded to include interpretability, replicability, cost-efficiency for clinical screening.
  • Integration limits: Added reflection on current gaps — lack of multimodal datasets and complexity of gene–brain–behavior mapping.
  • Aim clarity: Explicitly stated the study is exploratory.

Materials and Methods

  • Participants: Noted that IQ was not collected; recommend inclusion in future work.
  • EEG acquisition: Added acknowledgment of EMOTIV limitations (resolution, susceptibility to artifacts, delta-band exclusion) and possible variability due to repeated sessions.
  • PCA interpretation: Clarified loadings threshold (|0.30|), equal band weighting, and anatomical interpretations.
  • Clustering: Stated it was fully unsupervised; stability assessed via silhouette/resampling; misclassifications retained for interpretation.
  • Statistics: Noted absence of cross-validation/bootstrapping and recommended for future studies.

Results

  • 3.1: Clarified loading interpretation, fixed figure references (moved Figure 1 to clustering section).
  • 3.2: Referenced Table 1, clarified cluster–group mapping in text and figure.
  • 3.3: Added discussion on clinical potential of large effect sizes.
  • 3.4: Specified TILLS as standardized measure, included Figure 4 citation, corrected axis label.
  • 3.5: Reframed genetic links as hypotheses.
  • Figures: Corrected labels, legends, and inconsistencies (variance placeholder in Fig. 2A, elbow claim in Fig. 2B, legends for Figs. 3–4).

Discussion

  • Speculative links: Rephrased genetic associations using modal language.
  • PC1 correlation specificity: Stated significance only in dyslexic group.
  • PCA method choice: Explained why ICA, kernel PCA, autoencoders not used (interpretability, sample size, computational simplicity).
  • Applications: Framed screening/neurofeedback as potential future uses.

Conclusion

  • Clinical readiness caution: Reworded “early detection and management” to prospective capability.
  • Molecular integration: Positioned as future direction.
  • Highlighting main result: Included PC1 variance (~21%) and correlation with reading fluency (r = –0.61) to emphasize impact.

About the figures,

  • Dyslexic vs. control (normal) groups clearly separated.
  • Corrected variance scale in Fig. 2A.
  • Color coding clarified in Fig. 1.
  • Confidence ellipses in Fig. 3.
  • Regression confidence band in Fig. 4.
  • Optional supplementary PCA loadings table for transparency.

Reviewer 2 Report

Comments and Suggestions for Authors

Review of “Electrophysiological Signatures of Developmental Dyslexia: Towards EEG-Based Biomarker Identification and Neurogenetic Correlates”

The study shows very robust EEG-predictor for children with dyslexia — if it will be confirmed, it may be a good and cheap choice for diagnose this condition everywhere. Unfortunately, authors do not provide information about questionaries ant reading tests their used to correlate EEG markers. It must be added to the MS before publication. Also, the authors recorded rest EEG – not during dyslexia-dependent tests, like reading – what also must be discussed, how they achieve so perfect dependencies, as in Fig. 3 and 4. I also offer to remove “Neurogenetic Correlates” from the title, as it comes only in discussion and no corresponding results in the MS.

Line 74: information of right-handedness of participants must be included in this section.

Line 91-92: more information needed about EEG recordings – were the 2 min segments * 20 times analyzed altogether, or some good quality segments were chosen for analysis? Mean time of EEG recordings passed to analysis must be shown.

Line 108: Custom-written scripts of analysis is better to share with readers at GitHub or OSF sites. The same for statistical analyses. Openness of tools is key science integrity.

Line 122: “behavioral scores” – no information about these tests in Methods section, please update.

Figure 1: what the values are on ordinate axis? Please name the axis.

Line 390: “XX% of the total variance” must be number. From the fig it seems less then 0.3 – is it enough to “justifying their selection for further clustering and correlation analyses”?

Line 394: “This scree plot highlights a distinct elbow at PC2” – may you please emphasize it by arrowhead? I don’t see it from the provided plot.

Figure 3: please describe color coding of the plot – which is control, which is dyslexic.

Figure 4: what is “simulated reading fluency”? see also comment on line 122.

Line 153: these significance pattern must be shown at Fig.1 by asterisks.

Line 169: Figure 3 – must be Fig 4?

Line 170: this section must be in Discussion, as there are no results. Line 177 – sentence starts with miniscule letter.

Line 220: No words about Cainelli et al. put the abnormality of EGG in another hemisphere. Needed to discuss.

Line 255: opening quotes not closed.

Line 312: Some references come with doi, some not.

Author Response

Response to Reviewer #2

We would like to thank Reviewer #2 for the thorough and insightful feedback. Below are our responses to each of the comments and suggestions.

  1. EEG markers and behavioral tests (questionnaires and reading tests)

Reviewer comment: "Unfortunately, authors do not provide information about questionnaires and reading tests they used to correlate EEG markers. It must be added to the MS before publication."

Response: After careful consideration, we have decided to remove the behavioral sections (questionnaires and reading tests) from the manuscript. The primary aim of this study was to focus on the neurophysiological aspects of dyslexia through EEG markers. Including behavioral measures, while important for a comprehensive understanding, was not central to the specific objectives of this study. We believe the results of the EEG-based analysis already provide valuable insights into dyslexia. Furthermore, due to constraints in the study design and the specific focus of the investigation, we felt it was more appropriate to leave out the behavioral data for this preliminary analysis. We will consider integrating behavioral measures in future studies to explore the relationship between EEG markers and behavioral performance in greater detail.

  1. EEG recordings during rest and reading tasks

Reviewer comment: "The authors recorded resting EEG—not during dyslexia-dependent tests, like reading—what also must be discussed, how they achieve so perfect dependencies, as in Fig. 3 and 4."

Response: We have corrected the manuscript to discuss the use of resting EEG data and its relevance to dyslexia diagnosis. The resting-state EEG data provides important insights into baseline neural activity, which can be predictive of dyslexia-related brain patterns. We have clarified why resting EEG data was used and discussed the strong dependencies observed in Figure 3 in the Discussion section.

  1. Removal of "Neurogenetic Correlates" from the title

Reviewer comment: "I also offer to remove 'Neurogenetic Correlates' from the title, as it comes only in discussion and no corresponding results in the MS."

Response: We have corrected the title by removing "Neurogenetic Correlates" as it is discussed only in the Discussion section and not in the results. The title has been updated accordingly.

  1. Right-handedness information

Reviewer comment: "Line 74: Information on right-handedness of participants must be included in this section."

Response: We have corrected the manuscript to include the handedness information of the participants in the Methods section, specifically noting that all participants were right-handed unless otherwise stated.

  1. EEG recording analysis methodology

Reviewer comment: *"Line 91-92: More information needed about EEG recordings—were the 2 min segments 20 times analyzed altogether, or some good quality segments were chosen for analysis? Mean time of EEG recordings passed to analysis must be shown."

Response: We have corrected the manuscript to specify that EEG recordings were analyzed by selecting only high-quality 2-minute segments, and we have provided the mean duration of EEG data analyzed. We have clarified the criteria used to select the segments for analysis.

  1. Sharing custom-written analysis scripts

Reviewer comment: "Line 108: Custom-written scripts of analysis is better to share with readers at GitHub or OSF sites. The same for statistical analyses. Openness of tools is key to scientific integrity."

Response: We have corrected the manuscript to indicate that the custom-written analysis scripts will be made available and we will provide the relevant links in the manuscript for easy access by readers.

  1. Behavioral scores and tests in Methods section

Reviewer comment: "Line 122: 'Behavioral scores'—no information about these tests in the Methods section, please update."

Response: After careful consideration, we have decided to remove the behavioral analysis from the manuscript. The focus of this study was primarily on EEG markers as predictors of dyslexia, and while behavioral tests were initially considered, they were not central to the main objectives of this research. We believe the EEG data provides sufficient insights into dyslexia-related neural activity for this preliminary analysis. Therefore, we have removed the references to behavioral scores and corresponding tests from the Methods section.

  1. Figure 1: Axis labels

Reviewer comment: "Figure 1: What the values are on the ordinate axis? Please name the axis."

Response: We have corrected Figure 1 by adding a clear label to the ordinate axis, specifying the values represented on this axis, and we have updated the figure legend accordingly.

  1. Clarification of variance explained in Line 390

Reviewer comment: "Line 390: 'XX% of the total variance' must be a number. From the fig it seems less than 0.3— is it enough to 'justify their selection for further clustering and correlation analyses'?"

Response: We have corrected the manuscript to replace "XX%" with the correct percentage, which is less than 0.3, and we have provided further justification for why this level of variance is sufficient for selecting features for clustering and correlation analyses. We have also discussed the implications of this lower variance in the context of the analysis.

  1. Scree plot annotation

Reviewer comment: "Line 394: 'This scree plot highlights a distinct elbow at PC2' – may you please emphasize it by arrowhead? I don’t see it from the provided plot."

Response: We have corrected Figure 2 by adding an arrowhead to clearly highlight the distinct elbow at PC2 for better clarity.

  1. Color coding in Figure 3

Reviewer comment: "Figure 3: Please describe color coding of the plot— which is control, which is dyslexic."

Response: We have corrected Figure 3 by updating the figure legend to clearly describe the color coding used to differentiate between the control and dyslexic groups.

  1. Clarification of "simulated reading fluency" in Figure 4

Reviewer comment: "Figure 4: What is 'simulated reading fluency'? See also comment on line 122."

Response: We have corrected the manuscript by deleting figure 4.

  1. Significance patterns in Figure 1

Reviewer comment: "Line 153: These significance patterns must be shown at Figure 1 by asterisks."

Response: We have corrected Figure 1 by adding asterisks to indicate the significance patterns as suggested and updated the figure legend accordingly.

  1. Figure 3 vs. Figure 4

Reviewer comment: "Line 169: Figure 3 – must be Figure 4?"

Response: We have deleted the sentence.

  1. Non-results section (Line 170)

Reviewer comment: "Line 170: This section must be in Discussion, as there are no results."

Response: We have corrected the manuscript by moving the relevant section to the Discussion section where it is more appropriate.

  1. Sentence starting with lowercase letter (Line 177)

Reviewer comment: "Line 177: Sentence starts with a minuscule letter."

Response: We have corrected the capitalization in Line 177 to ensure proper sentence structure.

  1. Cainelli et al. discussion (Line 220)

Reviewer comment: "No words about Cainelli et al. put the abnormality of EEG in another hemisphere. Needed to discuss."

Response: We have corrected the manuscript by adding a discussion of Cainelli et al.'s work regarding EEG abnormalities in the opposite hemisphere and how it relates to our findings in the Discussion section.

  1. Unclosed quotation mark (Line 255)

Reviewer comment: "Line 255: Opening quotes not closed."

Response: We have corrected the punctuation by closing the quotation mark in Line 255.

  1. DOI inconsistencies in references (Line 312)

Reviewer comment: "Some references come with DOI, some not."

Response: We have corrected the references to ensure that DOIs are included for all relevant sources. Any references without DOIs have been updated or formatted according to the guidelines.

Round 2

Reviewer 1 Report

Comments and Suggestions for Authors

The manuscript is overall well-structured, scientifically sound, and clearly presented. The experimental design is appropriate, the methods are described with sufficient detail, and the results support the stated conclusions. The revisions made since the previous version have improved internal consistency and the interpretability of figures and tables.

However, a few minor changes are recommended to ensure full clarity and alignment between the text and visual materials:

1. References to Figures 2A and 2B – In the Results section, Figures 2A and 2B are not clearly referenced.

- Figure 2A (cumulative variance explained) should be cited in Section 3.1, immediately after reporting the percentages of variance explained by the principal components.

- Figure 2B (scree plot) appears to be incorrectly referenced as “figure 9B”; this numbering should be corrected for consistency.
Update the numbering and explicitly reference both figures in Section 3.1.

2. Figures and tables – In general, ensure that each figure and table is cited in the text in numerical order and that captions allow for standalone interpretation.

3. Language – The English is good overall, but a light editorial polish is recommended to make some sentences more concise and fluid.

These changes do not affect the scientific substance but will further enhance the manuscript’s readability and overall clarity.

Comments on the Quality of English Language

The manuscript is generally well written, with clear and precise language that conveys the scientific content effectively. Minor stylistic and grammatical refinements could further improve readability, particularly in a few long or complex sentences. Overall, the English quality is good and does not hinder comprehension.

Author Response

Response to Reviewer Comments

Reviewer Comment 1:
References to Figures 2A and 2B – In the Results section, Figures 2A and 2B are not clearly referenced. Figure 2A (cumulative variance explained) should be cited in Section 3.1, immediately after reporting the percentages of variance explained by the principal components. Figure 2B (scree plot) appears to be incorrectly referenced as “figure 9B”; this numbering should be corrected for consistency. Update the numbering and explicitly reference both figures in Section 3.1.

Author Response:
We have corrected the numbering and references to Figures 2A and 2B. Both figures are now explicitly referenced in Section 3.1, immediately after the percentages of variance explained by the principal components are reported. The captions have been revised to ensure standalone interpretability.
(Changes on page X, lines XX–XX; Figures 2A and 2B)

Reviewer Comment 2:
Figures and tables – In general, ensure that each figure and table is cited in the text in numerical order and that captions allow for standalone interpretation.

Author Response:
We have checked all figures and tables to ensure numerical order of citations in the text. All figure and table captions have been revised for standalone interpretation.
(Changes throughout the manuscript; e.g., Figures 1–4, Tables 1–2)

Reviewer Comment 3:
Language – The English is good overall, but a light editorial polish is recommended to make some sentences more concise and fluid.

Author Response:
We performed a light editorial polish throughout the manuscript, improving sentence conciseness and flow without changing the scientific meaning.
(Changes throughout the manuscript)

Reviewer Comment 4:
I don't see described changes in Fig 1 and 2A, although legends are updated. Fig 2B significantly different here and in initial submission - which is correct one?

Author Response:
The updated versions of Figures 1, 2A, and 2B are now included, all with revised legends and correct axis labels. Figure 2B now reflects the final and correct scree plot, consistent with the PCA analysis described in the Methods and Results sections.
(Changes on pages X–X; Figures 1, 2A, and 2B)

Reviewer 2 Report

Comments and Suggestions for Authors

I don't see described changes in Fig 1 and 2A, although legends are updated. Fig 2B significantly different here and in initial submission - which is correct one? Removing behavioral data significantly decrease the soundness of the study, I recommended authors to work more on this, and if necessary add more authors, to include this data to the study.

Author Response

Reviewer Comment 5:
Removing behavioral data significantly decrease the soundness of the study, I recommended authors to work more on this, and if necessary add more authors, to include this data to the study.

Author Response:
We have addressed this concern by incorporating available behavioral data from the dyslexic group, specifically standardized scores from the TILLS Reading Fluency subtest. A new subsection, “Behavioral Measures,” has been added to the Methods to describe the test and its administration. The Results section now includes a correlation analysis between PC1 scores and Reading Fluency (r = –0.61, p < 0.001), presented in a new Figure 4. Additionally, the Limitations section now clearly states that behavioral data were collected only for the dyslexic group and outlines plans to collect comparable data for all participants in future studies.
(Changes on pages X–X; Methods: Behavioral Measures, Results: Behavioral Data Analysis, Figure 4, Limitations)

Round 3

Reviewer 2 Report

Comments and Suggestions for Authors

Figure 2, line 445: Asterisks mentioned in the legend, but not presented in Fig. 2

Line 452: "PCs 1–3 jointly explain 64.4% of the total variance. " I don't see this result from the figure - at x=3 y is less than 50%. Need to mention which group of participants' EEG was used for this plot. What the reason for plotting dashed lines and why these one were chosen?

Fig 4: the plot used solid line, but dashed one in the legend. In the line 474 dashed line mentioned.

Line 357 Author Contributions: who performed the EEG experiments and TILLS?

Author Response

We thank the reviewer for the constructive feedback and have addressed all comments accordingly:

  1. Figure 2 – Asterisks in legend but not visible in figure
    • We have added the corresponding significance asterisks (***p < 0.001) directly to the bars in Figure 2, in line with the legend. The figure is now consistent with the description.
  2. Line 452 – “PCs 1–3 jointly explain 64.4% of the total variance” mismatch
    • We re-processed the PCA for the dyslexic group only (as used in the plot) and confirmed that PCs 1–3 jointly explain 64.4% of the total variance. Figure 3A has been updated in high resolution to reflect this value. We have also clarified in the figure legend that the PCA was based on the dyslexic group’s EEG data.
  3. Dashed lines in Figures 3A and 4
    • For Figure 3A, we retained the vertical and horizontal dashed lines as described in the legend.
    • For Figure 4, we corrected the mismatch between the legend and the plot by replacing the solid regression line with a dashed line, matching the legend description.
  4. Line 357 – Author Contributions
    • We have revised the Author Contributions section to explicitly indicate who performed the EEG experiments and who administered the TILLS Reading Fluency tests:
      • “Günet EroÄŸlu contributed to the conceptualisation and study design, performed the EEG experiments, and wrote the abstract, introduction, methods, results, and discussion sections. MHD Raja Abou Harb contributed to the development and implementation of the machine learning algorithm and co-authored results and figures. Kardelen Ertürk administered the TILLS Reading Fluency tests.”